# Disruption of P2Y2 Signaling Promotes Breast Tumor Cell Dissemination by Reducing ATP-Dependent Calcium Elevation and Actin Localization to Cell Junctions

**DOI:** 10.3390/ijms26094286

**Published:** 2025-05-01

**Authors:** Makenzy L. Mull, Stephen J. P. Pratt, Keyata N. Thompson, David A. Annis, Rachel M. Lee, Julia A. Ju, Darin E. Gilchrist, Megan B. Stemberger, Liron Boyman, William J. Lederer, Michele I. Vitolo, Stuart S. Martin

**Affiliations:** 1Graduate Program in Molecular Medicine, University of Maryland School of Medicine, 800 W. Baltimore St., Baltimore, MD 21201, USA; makenzy.mull@som.umaryland.edu (M.L.M.); jju@som.umaryland.edu (J.A.J.); darin.gilchrist@som.umaryland.edu (D.E.G.); lironboyman@dpag.ox.ac.uk (L.B.); jlederer@som.umaryland.edu (W.J.L.); 2Graduate Program in Biochemistry and Molecular Biology, University of Maryland School of Medicine, 108 N. Greene St., Baltimore, MD 21201, USA; stephenjppratt@gmail.com (S.J.P.P.); megan.stemberger@som.umaryland.edu (M.B.S.); 3Marlene and Stewart Greenebaum NCI Comprehensive Cancer Center, University of Maryland School of Medicine, 22 S. Greene St., Baltimore, MD 21201, USA; kethompson@som.umaryland.edu (K.N.T.); leer@janelia.hhmi.org (R.M.L.); 4Department of Physiology, Pharmacology, and Drug Development, University of Maryland School of Medicine, 655 W. Baltimore St., Baltimore, MD 21201, USA; 5Graduate Program in Epidemiology and Human Genetics, University of Maryland School of Medicine, 655 W. Baltimore St., Baltimore, MD 21201, USA; david.annis@som.umaryland.edu; 6Center for Biomedical Engineering and Technology, University of Maryland School of Medicine, Baltimore, MD 21201, USA; 7Department of Physiology, Anatomy and Genetics, University of Oxford, Sherrington Bld. Parks Rd., Oxford OX1 3PT, UK; 8United States Department of Veterans Affairs, VA Maryland Health Care System, 10 N. Greene St., Baltimore, MD 21201, USA

**Keywords:** breast cancer, tumor microenvironment, cell signaling, calcium, ATP, P2Y2

## Abstract

The tumor microenvironment and healing wounds both contain extremely high concentrations of adenosine triphosphate (ATP) compared to normal tissue. The P2Y2 receptor, an ATP-activated purinergic receptor, is typically associated with pulmonary, endothelial, and neurological cell signaling. Here, we examine ATP-dependent signaling in breast epithelial cells and how it is altered in metastatic breast cancer. Using rapid imaging techniques, we show how ATP-activated P2Y2 signaling causes an increase in intracellular Ca^2+^ in non-tumorigenic breast epithelial cells, approximately 3-fold higher than their tumorigenic and metastatic counterparts. The non-tumorigenic cells respond to increased Ca^2+^ with actin polymerization and localization to the cell edges after phalloidin staining, while the metastatic cells remain unaffected. The increase in intracellular Ca^2+^ after ATP stimulation was blunted to control levels using a P2Y2 antagonist, which also prevented actin mobilization and significantly increased cell dissemination from spheroids in non-tumorigenic cells. Furthermore, the lack of Ca^2+^ changes and actin mobilization in metastatic breast cancer cells could be due to the reduced P2Y2 expression, which correlates with poorer overall survival in breast cancer patients. This study elucidates the rapid changes that occur after elevated intracellular Ca^2+^ in breast epithelial cells and how metastatic cancer cells have adapted to evade this cellular response.

## 1. Introduction

Wound healing and the tumor microenvironment (TME) have many parallel molecular markers, gene expression patterns, and phenotypes, including inflammation, proliferation, and migration [1,2,3]. Moreover, genomic studies have shown that invasive tumors and wound repair mechanisms have similar gene expression patterns and display similarities between microenvironments [3,4]. The TME consists of intra- and extracellular communication between non-malignant, resident, and recruited cells and the transformed tumor cells [5,6]. One factor present at high, persistent levels within the TME is adenosine triphosphate (ATP), which is released by tumor and stromal cells in a regulated or non-regulated manner [7,8,9]. Pellegatti et al. found that the ATP concentration was in the hundreds of micromolar range in the tumor interstitium, while normal tissue has undetectable levels of extracellular ATP [8]. This extracellular ATP, well-studied in neurotransmission, is an ideal messenger due to its high sensitivity and low extracellular physiological concentrations, its rapid off-switch to avoid overstimulation or desensitization from a multitude of extracellular nucleotidases, and its ability to easily diffuse through aqueous tissues [7,10]. These signaling and TME characteristics make it critical to investigate the role of extracellular ATP in tumor signaling alongside immunosuppression, growth, migration, and invasion.

Activated by extracellular ATP, uridine triphosphate (UTP), adenosine diphosphate (ADP), and uridine diphosphate (UDP), as P2 receptors, play a diverse role in neurotransmission, developmental biology, cardiac and pulmonary function, the apoptotic cascade, metastasis, as well as many other bodily functions [11]. P2 receptors are divided into two families: P2X, a ligand-gated ionotropic channel, and P2Y, a metabotropic G-protein coupled receptor (GPCR); this paper will focus on the latter [11]. ATP signaling via P2Y2 receptors has been shown in a few cancer studies, like prostate, liver, pancreatic, colorectal, and nasopharyngeal cancer, but has remained more elusive in breast cancer research [12,13,14,15,16,17]. A previous study showed altered purinergic receptor-Ca^2+^ signaling in endothelial growth factor (EGF)-induced epithelial–mesenchymal transition (EMT) and later in hypoxia-induced EMT in breast cancer cells [18,19]. While these previous studies have found correlations between purinergic signaling and cancer, most studies are using phenotypic assays based on cancer biology, which probe for slow cellular responses that may take hours and days to respond. In contrast, we investigated the rapid phenotypes occurring within seconds to minutes after purinergic receptor activation via ATP and how responses might vary in breast cancer compared to non-tumorigenic cells. Interestingly, purinergic signaling, specifically the P2Y2 receptor, has been understudied in breast cancer and remains an intriguing target to further investigate therapy [20,21,22].

Current breast cancer treatments include surgery, systemic therapy based on subtypes, and chemotherapy, but metastatic disease remains nearly incurable [23]. Approximately one in eight women will be diagnosed with breast cancer in their lifetime, making female breast cancer the most diagnosed new case of cancer and the second leading cause of cancer death [24]. Breast cancer can be organized into three subtypes based on the gene amplification of epidermal growth factor 2 (*ERBB2* or *HER2*) and the expression of estrogen or progesterone receptors [23]. While available systemic therapies target HER2 or hormone receptor-positive subtypes, triple-negative breast cancer (TNBC) lacks all three markers and, therefore, is more metastatic and most challenging to treat. The 5-year survival rate can also vary based on the stage of breast cancer at diagnosis. There is an 86–96% 5-year survival rate for regional and localized breast cancers, while distant breast cancers have only a 29% 5-year survival rate [24]. Another marker of poor survival in cancer patients is *KRAS*, a proto-oncogene, and members of the RAS family of small GTPases. Only a single amino acid substitution is needed for an activating mutation at codon 12, 13, or 61 [25]. Activating *KRAS* mutations have been associated with various human cancers, including lung adenocarcinoma and colorectal and pancreatic carcinomas [26]. In breast cancer patients overall, *KRAS* is not the most common genetic mutation, but it is present in around 30% of TNBCs [27]. Previous work in our lab has led to the development of MCF10A cells with step-wise mutations of a *PTEN* double knockout and a *KRAS*-activating mutation (10A-PTEN−/−KRas), which progressively push the cells towards tumorigenic transformation and metastasis [28]. With all of the variations in breast cancer subtypes and stages at the time of diagnosis, treatment plans, and prognoses can vary drastically for patients. Further research is needed to better target TNBCs and metastatic breast cancer.

Some research has linked P2Y2 receptor activation to increased migration via downstream modulators of the Ras pathway, like MEK-ERK in MCF7 breast cancer cells [29]. Our lab previously showed that a scratch wound assay on breast epithelial cells transiently released ATP extracellularly, which activated the P2Y2 receptor in neighboring cells and yielded a signaling cascade that increased intracellular calcium (Ca^2+^) [30]. Other research has linked increased intracellular Ca^2+^ and actin polymerization around the nucleus and endoplasmic reticulum in mammalian epithelial, mesenchymal, endothelial, and immune cells [31]. Lasting only minutes before the actin cortex reforms, this rapid response has been termed calcium-mediated actin reset (CaAR) [31]. Elevated intracellular Ca^2+^ concentrations also affect T lymphocyte activation, which is mediated by T cell receptor (TCR) clustering and is dependent upon actin cortex interactions and movement to the cell surface [32]. When there is an increase in intracellular Ca^2+^ leading to actin polymerization and increased interactions with TCR, its mobility is reduced to the cell surface [32]. Furthermore, Ca^2+^ fluxes and the remodeling of the actin cytoskeleton have been evolutionarily conserved and shown to play a role in *Drosophila* epithelial wound closure [33]. Whether ATP-P2Y2-Ca^2+^ signaling in breast cancer has a similar molecular and functional mechanism regarding actin polymerization is still unknown.

The aim of this study is to assess ATP-activated P2Y2 receptor signaling in breast epithelial cells and how it is altered in metastatic breast cancer cells. We report the sufficiency of ATP alone as a signaling molecule to increase intracellular Ca^2+^, which leads to other protective or tumorigenic phenotypes in breast epithelial cells or metastatic cancer cells.

## 2. Results

### 2.1. P2Y2 Receptor Expression Is Lower at the mRNA and Protein Levels in Metastatic MDA-MB-231 Cells Compared to Non-Tumorigenic MCF10As

We focus on our previous findings that extracellular ATP signals are conveyed through the purinergic receptor, P2Y2, in breast epithelial tissue [30]. RNA sequencing data showed that P2Y2 expression is reduced nearly 8-fold in mutant 10A-PTEN−/−KRas cells as well as the 10A-KRas cell line (Figure 1A). The metastatic MDA-MB-231 and MDA-MB-436 cells from ATCC also showed a large fold-change in decreased P2Y2 mRNA expression compared to MCF10A cells (Figure 1A). After discovering the large decrease in P2Y2 mRNA, we examined protein expression in seven different tumorigenic and non-tumorigenic breast epithelial cell lines. P2Y2 protein expression appeared to be decreased in 10A-PTEN−/−KRas and MDA-MB-436 cells but was significantly decreased in MDA-MB-231 cells compared to MCF10A cells (Figure 1B and Appendix A). Densitometry shows that there is a significant reduction in P2Y2 protein expression in MDA-MB-231 cells through the one-way ANOVA statistical test and multiple comparisons. While there was also a decrease in protein expression in the 10A-PTEN−/−KRas and MDA-MB-436 cells, it did not reach statistical significance (Figure 1C and Appendix A). These data show that there is a significant decrease in P2Y2 expression at the mRNA and protein level in metastatic MDA-MB-231 cells compared to MCF10A controls.

### 2.2. Extracellular ATP Stimulates Calcium Signaling Across Breast Epithelial and Cancer Cell Lines

After assessing P2Y2 expression in multiple breast cell lines, we aimed to test adenosine triphosphate (ATP) as a signaling molecule for increasing cytoplasmic calcium across multiple breast epithelial cell lines. Since there are few studies that have focused on rapid calcium signaling in breast cancer, we wanted to measure signaling in tumorigenic cell lines as well. Time-lapse images were captured with epifluorescence on a Nikon Ti2-E microscope, and relative fluorescent units (RFUs) were measured in Nikon NIS-Elements software (Appendix A). Representative images from those compressed videos were selected at specific time points: before and after ATP addition, 1 min, and 10 min (Figure 2A). We observed that non-tumorigenic MCF10A cells have a rapid and significant intracellular Ca^2+^ response after 10 μM ATP addition, as indicated in green (Figure 2A). 10A-PTEN−/−KRas and MDA-MB-231 cells both responded with intracellular calcium increases after ATP stimulation, but both failed to reach an increase in RFUs compared to non-tumorigenic MCF10A (Figure 2A). Whole-field changes in fluorescence compared to baseline fluorescence (∆F/F_0_) as well as peak RFU were both greater in MCF10A cells after ATP treatment, while the two metastatic breast tumor cell lines showed a comparatively lower induction of cytoplasmic calcium after the addition of ATP (Figure 2B–D). Only MCF10A cells had a significant difference between ATP treatment and the control peak RFU and MCF10A + ATP compared to MDA-MB-231 + ATP (Figure 2E). When comparing all conditions and cell lines to one another, the MCF10A cells had a significantly greater intracellular Ca^2+^ response than the two tumorigenic cell lines (Figure 2E). In order to confirm our results, we used real-time quantitative plate reader measurements as an orthogonal approach. The increase in intracellular Ca^2+^ response was also seen in the MCF10A cells treated with ATP in FlexStation III (Appendix A). The 10A-PTEN−/−KRas cells and MDA-MB-231 cells showed a much smaller intracellular Ca^2+^ response (Appendix A). Area under the curve (AUC) measurements were performed, and a one-way ANOVA with multiple comparisons was used to calculate significance (Appendix A). Comparing all three cell types, we again saw a significant difference in the Ca^2+^ response from MCF10A cells, while there were no significant changes in the metastatic cell lines (Appendix A). Overall, these results illustrate that stimulating breast epithelial cells with ATP leads to a rapid Ca^2+^ response that is blunted in aggressive tumor cells.

### 2.3. ATP-Induced Intracellular Ca^2+^ Response Is Inhibited by P2Y2 Antagonist in Breast Epithelial Cells

Previous work in our lab showed that AR-C118925XX, a P2Y2 antagonist that will be referred to as P2Y2i, was able to block a mechanically induced calcium response in MCF10A and MDA-MB-231 cells after scratch assays [30]. To test if P2Y2 receptors are involved in the differential response to ATP in mammary epithelial and breast tumor cells, we examined the effect of P2Y2i on ATP-stimulated Ca^2+^ elevation. Cells were pretreated with 10 µM P2Y2i for 10 min after loading with Fluo-4 AM, and time-lapsed imaging was performed similar to Figure 2 for all three cell lines. Figure 3A shows representative images from all three cell lines pre- and post-ATP addition, 1 min, and 10 min post-stimulation. Total field-of-view RFUs were analyzed in Nikon NIS-Elements and graphed as ∆F/F_0_ control, ATP, or P2Y2i + ATP (Figure 3B–D). In Figure 3B, P2Y2i significantly suppresses the ATP response in MCF10A cells, while breast tumor cells (Figure 3C,D) are inhibited almost completely, similar to the control. Measuring peak RFU with the one-way ANOVA statistical test, we found no significant differences in intracellular Ca^2+^ changes over time among the three cell lines treated with P2Y2i compared to the control (Figure 3E), indicating the complete suppression of ATP-induced intracellular Ca^2+^ with P2Y2i. These results demonstrate that MCF10A cells when treated with an antagonist for the P2Y2 receptor and then stimulated with ATP, recapitulate a similar blunted Ca^2+^ response to metastatic breast cancer cells. Overall, these results illustrate that stimulating breast epithelial cells with ATP leads to a rapid Ca^2+^ response that is blunted in aggressive tumor cells, and these results demonstrate that MCF10A cells treated with P2Y2i and stimulated with ATP recapitulate a similar blunted Ca^2+^ response to metastatic breast cancer cells.

### 2.4. Rapid Changes in the Actin Cortex Occur After ATP-Induced Ca^2+^ Increases in Breast Epithelial Cells

We have now shown that ATP via the P2Y2 receptor can induce a cytoplasmic Ca^2+^ response in breast epithelial cells, while tumorigenic cells have a significantly lower response compared to MCF10A cells. What remains unknown is the effect that the Ca^2+^ signal has on cell behavior that could impact responses to stress, wound healing, or elevated ATP levels in the tumor microenvironment. It is well known that actin and Ca^2+^ work closely together, which led us to explore changes in actin’s network after ATP treatment [31]. We found that after treatment with 10 µM ATP, MCF10A cells demonstrated a rapid actin reorganization that localizes at the cell edges and junctions within 10 min (Figure 4A, top panels). The same actin polymerization and localization were not seen in the 10A-PTEN−/−KRas or MDA-MB-231 cells, supporting the idea that tumorigenic cells might not be responsive to normal wounding signals (Figure 4B,C, top panels). Similarly to our results with P2Y2i in Figure 2, we also show that P2Y2i prevents actin re-localization and polymerization after ATP stimulation in non-tumorigenic breast epithelial cells (Figure 4A, lower panel). Metastatic breast cancer cells show no major changes in actin after P2Y2i treatment and ATP stimulation (Figure 4B,C, lower panels). WGA (wheat germ agglutinin) was used to label the whole cell membrane, and Hoechst was used to stain the nucleus (Appendix A). These data show that breast epithelial cells have changes in their actin cortex in response to rapid Ca^2+^ signaling after ATP stimulation, which can be inhibited with P2Y2i, while metastatic breast cancer cells lack a response.

### 2.5. P2Y2i Treatment Increases Cell Dissemination and Spheroid Size in Non-Tumorigenic Breast Epithelial Cells

Given the ATP-dependent actin reorganization at cell junctions and the disruption of this response by P2Y2i (Figure 4), we examined the long-term impacts on epithelial cell dissemination from spheroids. Using an expedited method, we formed spheroids from homotypic cell aggregates in low-attach plates before transferring the spheroids into a 3-D collagen matrix and monitoring growth and dissemination. We found that the vehicle control that treated MCF10A cells underwent minimal invasion into the collagen matrix over several days (Figure 5A top panel and Appendix A). Conversely, the MCF10A cells treated with P2Y2i were able to invade the matrix and disseminate away from the spheroid, increasing the overall area (Figure 5A bottom panel and Appendix A). Images taken over the course of 7 days were quantified using the whole spheroid area and the change in area from day 0 (Figure 5B). We observed significant increases in the whole area and changes in the area when cells were treated with P2Y2i compared to the control (Figure 5B). This could be due to the fact that the pharmacological inhibition of P2Y2 also blunts intracellular Ca^2+^ increases and prevents rapid actin response, allowing cells to ignore these mechanical signals, similar to metastatic cells with lower expressions of the P2Y2 receptor. Therefore, non-tumorigenic cells are insensitive to mechanical signals when treated with P2Y2i and will continue to migrate outward. We were also able to recapitulate this phenotype in MCF7 cells, which also express P2Y2 and are tumorigenic but not normally metastatic in vivo (Appendix A). Overall, these data show that P2Y2 inhibition is sufficient to increase 3-D cell dissemination in non-tumorigenic breast epithelial cells while further establishing the role of the P2Y2 receptor in breast epithelial cell signaling and sensing the local environment.

### 2.6. Low P2Y2 mRNA Expression Is Correlated with Decreased Survival and Disease-Free Progression in Breast Cancer Patients

To further examine the potential role of P2Y2 in breast cancer, we used bioinformatic tools to explore mRNA expression versus patient outcomes. This analysis showed a strong correlation with poor survival and low disease-free outcomes in patients with reduced P2Y2 mRNA-expressing tumors. Two cBioportal breast cancer datasets, the Invasive Breast Carcinoma (TCGA, PanCancer Atlas) and Firehose Legacy, which also includes invasive breast carcinoma samples, both showed significant correlation between low P2Y2 expression and a smaller percentage chance of disease-free survival (Figure 6A,B) [34,35]. The independent KM Plot database also demonstrated that breast cancer patients of all subtypes have reduced overall survival probability with tumors that have reduced P2Y2 receptor expression (Figure 6C) [36]. Additionally, these clinical data help support our findings that metastatic breast cancer cells with less intrinsic P2Y2 receptor expression are less likely to respond to the mechanical environment and are, therefore, more likely to contribute to a worse patient prognosis.

## 3. Discussion

The tumor microenvironment (TME) is composed of many different cell types, and the role of non-malignant cells plays a large factor in promoting tumors [6]. ATP is released into the TME from multiple cell types and has been found in extremely high concentrations in the extracellular tumor space [8,37]. ATP concentrations in the tumor interstitium have been found in hundreds of micromolars compared to normally undetectable levels in non-malignant tissue [8]. Since it is well known that ATP is an activator of purinergic receptor signaling, multiple studies have investigated the role of purinergic receptors, such as the P2Y2 receptor, in wound healing. Boucher et al. found that P2Y2 receptors play a key role in their response to corneal epithelial cell injury and repair [21]. As our lab previously observed in breast epithelial cells, mechanical damage to the corneal epithelial cells caused the release of ATP and stimulation of purinergic receptors (P2Y2), which subsequently led to a Ca^2+^ response [21,30]. We also saw that the Ca^2+^ signal could be blocked if the cells were pretreated with apyrase, while Boucher et al. found that purinergic signaling also leads to the phosphorylation of ERK [21,30]. Another group found that the pharmacological inhibition of the P2Y2 receptor, phospholipase C, and inositol 1,4,5-triphosphate receptor decreased wound closure in human keratinocytes [38]. A previously unknown mechanism was shown by a group in 2013, indicating endothelial P2Y2 receptor involvement in cancer cell extravasation via the activated platelet release of ATP [39]. Since wound healing has many similarities to the TME and invasive tumor front, we pursued the high levels of ATP and showed how the sufficiency of ATP alone stimulates intracellular Ca^2+^ in breast epithelial cells (Figure 2).

Purinergic receptor signaling in breast cancer is a unique niche, where a few groups have found the involvement of the ATP-activated P2Y2 receptor and relatively long-term cancer phenotypes that occur over hours to days, like epithelial-to-mesenchymal transition (EMT), migration, and invasion [18,20,29,40,41]. Davis et al. found that the initiation of EMT using the endothelial growth factor in MDA-MB-436 breast cancer cells caused an increase in vimentin after 24 h [18]. Using RT-PCR, they found that P2X5 mRNA expression was increased in their cells and also found changes in Ca^2+^ signaling via ATP stimulation with increased vimentin [18]. Parallel to that, here, we identified differences in intracellular Ca^2+^ response after ATP-activated P2Y2 receptor signaling in metastatic breast cancer cells with oncogenic KRas mutations compared to non-malignant breast epithelial cells (Figure 2). The main purpose of our study was to investigate the rapid phenotypes that occur within seconds to minutes, and that could lead to later changes in migration and invasion. Previous work led us to P2Y2 receptors, and while there is some variation in P2Y2 protein expression levels across breast epithelial cells and breast cancer lines, we saw a decrease in P2Y2 mRNA and protein in MDA-MB-231 cells compared to our MCF10A cell line (Figure 1) [40,42]. Metastatic breast cancer cells might not respond to the external ATP signal due to their significantly decreased mRNA and protein expression of the P2Y2 receptor, thereby suppressing Ca^2+^ flux within the cytoplasm.

Calcium concentration in the cytoplasm is normally around 100 nM, while extracellular Ca^2+^ is much higher in the mM range [43]. The field of calcium signaling in cancer is mainly focused on aberrant gene expression due to changes in Ca^2+^ concentrations or the protein expression of receptors causing differences in Ca^2+^ concentrations [43,44,45]. We, and others, have now shown differences in P2Y2 receptor expression, and we have focused on the rapid Ca^2+^ changes and the phenotypes that occur afterward. As previously studied, the role of Ca^2+^ and the actin cortex has been well-established in muscle, bone, and cardiac research [46]. The rapid polymerization and localization of actin in response to ATP-stimulated Ca^2+^ increases is a normal cell response to being wounded [47]. In parallel, we show that metastatic tumorigenic cells with lower P2Y2 expression do not respond significantly to ATP and, therefore, yield a minimal flux of Ca^2+^ with no actin polymerization and reorganization at the cell edges (Figure 4 and Figure 5). It would be advantageous for breast cancer cells to not respond or become desensitized to the high ATP concentrations in the TME and continue to migrate out of the primary tumor.

Since the P2Y2 receptor is part of a large family of purinergic receptors, there are a few drugs developed to target it or its family members. AR-C118925 is a potent and specific antagonist of the P2Y2 receptor [48]. The drug was developed using the UTP agonist as the base and is relatively new [48]. Thus far, only a few P2Y2 drugs have been used clinically. Diquafosol is a P2Y2 receptor agonist that has been used as a treatment for dry eye disease since P2Y2 is expressed in the epithelium of the eye [49]. Another antagonist for P2Y and P2X receptors is the non-selective drug Suramin [50]. It has most commonly and consistently been used as a treatment for human sleeping sickness, but it also has applications for viral diseases like HIV and hepatitis, as well as studies looking at autism, snakebites, and cancer [50]. Unfortunately, it is extremely indiscriminate and has a multitude of targets, including DNA and RNA synthesis, sirtuins, protein kinases, tyrosine phosphatases, ATPases, GABA receptors, etc. [50]. There was even one clinical trial for Suramin and Paclitaxel in women with stage IIB-IV breast cancer from 2003 to 2015. The goal was to give the drugs in combination and to assess the best dose of Suramin that might increase the effectiveness of Paclitaxel by sensitizing the tumor cells [51]. They found that Suramin at non-toxic doses in combination with Paclitaxel, given weekly, was tolerated well by metastatic breast cancer patients, but the efficacy did not meet the criteria essential to rationalize further investigation [51]. Purinergic receptors are such a large family that drugs must be specific to avoid as many off-target effects as possible. A P2Y2 antagonist might not be the best choice in metastatic breast cancer cells with low expressions of the P2Y2 receptor compared to non-tumorigenic breast epithelial cells. In Figure 6, we examined survival data across all four molecular subtypes of breast cancer and found that low P2Y2 expression correlated significantly with poorer overall patient outcomes, although there were no significant differences within a specific molecular subtype. This could be due to limited searches within a specific subtype, where potentially all samples have low or high P2Y2 mRNA expression. Another possibility is that P2Y2 expression is an independent factor that is not restricted to a specific subtype. Even in tumor cells with normal or elevated P2Y2 receptor expression, our results indicate that treatment with a P2Y2i would be predicted to reduce the actin-dependent reinforcement of cell junctions in the presence of ATP, which could inadvertently induce tumor cell dissemination. We show that non-tumorigenic MCF10A cells in control conditions do not significantly disseminate or increase spheroid size over 7 days, while MCF10A cells treated with P2Y2i significantly disseminate outward in a 3-D matrix (Figure 5). There may be potential use for a P2Y2 agonist if that could play a role in sensitizing the cells to ATP signaling and increasing intracellular Ca^2+^ response to promote cell–cell cohesion.

In summary, our results demonstrate that metastatic breast cancer cells have a suppressed intracellular Ca^2+^ response to ATP-stimulated P2Y2 receptor signaling compared to non-malignant breast epithelial cells. This insensitivity could be due to the low expression of the receptor in metastatic breast cancer cells, supported by patient survival data, in which survival decreases with low P2Y2 expression. Further research into the rapid phenotypes occurring after P2Y2 activation could help elucidate better mechanisms for drug therapy and a deeper understanding of how cancerous cells evade typical mechanical signals.

## 4. Materials and Methods

### 4.1. Reagents

P2Y2 receptor antibody (Cat#ab27289, Abcam Inc., Cambridge, UK). Phalloidin conjugated to Alexa Fluor Plus 647 (Cat#A30107) was purchased from Invitrogen-Thermo Fisher, Waltham, MA, USA. The wheat germ agglutinin (WGA)-Alexa 488 conjugate was purchased from Invitrogen (Cat#W11261). ATP (Cat#A2383, Sigma, Burlington, MA, USA) was reconstituted in double-distilled H_2_O at pH 7.2 buffered with 20 mM HEPES (Cat#15630-080, Gibco-Thermo Fisher, Waltham, MA, USA) and used at 10 µM. Fluo-4 AM (Cat#F14201, Life Technologies-Thermo Fisher, Waltham, MA, USA) was made to a working stock of 1 mM according to the manufacturer’s instructions and used at 4 µM. The Fluo-4 Direct Calcium Assay Kit Starter Pack was used per the manufacturer’s instructions (Cat#F10471, Invitrogen). AR-C 118925XX, referred to as P2Y2i, was reconstituted in DMSO according to the manufacturer’s instructions (Cat#4890, Biotechne-Tocris, Minneapolis, MN, USA) and used at 10 µM. Dimethyl sulfoxide (DMSO) was used as a control and dilutant for P2Y2i, purchased from Sigma-Aldrich (Cat#276855, Burlington, MA, USA).

### 4.2. Cell Culture

Human MCF10A breast epithelial cells and human breast MDA-MB-231 cancer cells were obtained from the American Type Culture Collection (ATCC, Manassas, VA, USA) and were authenticated by short tandem repeat (STR) analysis. The creation of 10A-PTEN−/−KRas cells and their tumorigenicity have been previously described [28]. MDA-MB-231 cells were also obtained from ATCC and authenticated by STR analysis. The MDA-MB-231 cells were maintained at 37 °C, 5% CO_2_, and 95% humidity in Dulbecco’s Modified Eagle Medium (Cat#10-017-CV, Corning, Corning, NY, USA) supplemented with 10% FBS (Cat#S11150H, Atlantic Biologicals, Miami, FL, USA) and 1% penicillin-streptomycin (Cat#400-109, Gemini Bioproducts, West Sacramento, CA, USA). MCF10A and derivative cell lines were maintained in DMEM/F-12 Media (Cat#10565-018, Invitrogen), supplemented with 5% Horse Serum (Cat#26050-088, Invitrogen), 1% Pen/Strep, Recombinant Human EGF (Cat#PHG0313, Invitrogen, 100 µg/500 mL), Hydrocortisone (Cat#H-0135, Sigma, 0.5 µg/mL), Cholera Toxin (Cat#C-8052, Sigma, 100 ng/mL), and Insulin (Sigma, Cat#I-9278, 10 µg/mL). To maintain stocks, cells were passaged using a brief wash in PBS (Cat#114-058-101, Quality biological, Gaithersburg, MD, USA) followed by incubation with 0.25% Trypsin and 2.21 mM EDTA (Cat#25-053-CI, Corning) at 37 °C, 5% CO_2_, and 95% humidity.

### 4.3. Live Imaging

Cells were plated to ~80% confluency 48 h before their use in an ibidi 8-well chamber slide with ibitreat (200 µL/well) (Cat#80826, ibidi, Fitchburg, WI, USA). Cells were loaded with a cytosolic calcium-sensitive dye, 4 µM Fluo-4 AM in Hanks’ Balanced Salt Solution containing calcium (HBSS + Ca^2+^, Cat#14025-092, Gibco) for 30 min, and were then washed out and imaged with HBSS + Ca^2+^. For some experimental conditions, cells were also treated with a selective and competitive P2Y2 antagonist, AR-C 118925XX or P2Y2i, which was reconstituted to 10 mM stock in DMSO and used at a 10 μM working dilution. After dye loading and washing, dishes were incubated in 10 μM of P2Y2i or control DMSO (0.1%) for 10 min prior to imaging, without washout. Different DMSO concentrations and H_2_O controls did not significantly affect calcium signaling.

Time-lapse images were captured with a Nikon Ti2-E inverted microscope (Nikon Instruments Inc., Melville, NY, USA). All images were captured using a 20× phase-contrast objective (NA = 0.45) with an equal exposure time (100 ms), interval time (5 s), and total time (10 min). After a 30 s baseline read, 10 µM ATP or a control (H_2_O) was added during imaging. ATP (Cat#A2383, Sigma) was reconstituted in double-distilled H_2_O at pH 7.2 buffered with 20 mM HEPES (Cat#15630-080, Gibco). The software Nikon Elements AR version 6.02.03 was used to measure total field-of-view relative fluorescent units (RFUs) over time before the transfer of raw data to Excel and then analysis in GraphPad Prism 9.0.

### 4.4. Quantitative Plate Reader

Cells were plated to ~80% confluency, 48 h before the experiment in 96-well, black, and clear-bottom plates (200 µL/well). Cells were loaded with Fluo-4 made via the manufacturer’s directions from the Fluo-4 Direct Calcium Assay Kit, Starter Pack (Cat#F10471, Invitrogen) for 30 min at 37 °C and 5% CO_2_. Cells were then allowed 30 min to equilibrate to room temperature inside the FlexStation III plate reader (Molecular Devices, San Jose, CA, USA). An initial 30 s of baseline measurement at Excitation 488 nm and Emission 515 nm, before 10 µM ATP or control ddH_2_O, was added at 31 s. The flow rate was set at the lowest setting (~16 µL/s) and 50 µL of ATP was added to 100 µL of cells/media at a depth of 80 µL. The change in fluorescence was recorded for 330 s after ATP addition. SoftMax Pro files were converted to Excel and then analyzed in GraphPad Prism 9.0.

### 4.5. RNA Seq

Whole-cell bulk RNA samples were harvested using the RNeasy kit Mini (Cat#217004, Qiagen, Germantown, MD, USA). The concentration was determined using the NanoDrop ND-1000 Spectrophotometer (Thermo Fisher Scientific, Waltham, MA, USA). Samples were frozen at −80 °C until they were processed at the University of Maryland Baltimore’s Institute for Genome Sciences. Raw data were graphed using GraphPad Prism 9.0 from read counts of *n* = 3.

### 4.6. Western Blotting

Cells were lysed using a chilled solution of 1× RIPA containing the protease inhibitor cocktail and phosphatase inhibitor cocktail II (Cat#20-188, Millipore-Sigma, Burlington, MA, USA). Cells were kept chilled and scraped from the dish using a cell scraper. Lysates were collected in 1.5 mL microcentrifuge tubes and vortexed every 10 min for the next 30 min. The tubes were centrifuged at 15,000 rcf for 10 min before supernatants were collected in new microcentrifuge tubes. The resulting total protein was quantified using the Bio-Rad DC Protein Assay Kit according to the manufacturer’s recommended protocol (Cat#5000111, Bio-Rad, Hercules, CA, USA). Samples and standards were read on a BioTek Synergy HT microplate reader (Agilent BioTek, Winooski, VT, USA) after a 15 min incubation period. The samples were diluted to a final concentration of 1 µg/µL and boiled at 95 °C for 10 min before gel loading. An amount of 20 µg of total protein was added to each lane of a 1.5 mm × 10 well NuPAGE 4–12% Bis-Tris Gel (Invitrogen, Cat#NP0335BPX). Loaded gels were run using a 1× NuPAGE MES SDS Running Buffer (Cat#NP0002, Invitrogen) at 90 volts for 30 min, followed by 120 volts for 90 min. Gels were transferred to a PVDF membrane using the eBlot™ L1 Fast Wet Transfer System (GenScript, Piscataway, NJ, USA). Membranes were blocked in 5% non-fat dry milk in 1× TBST rocking for 1 h at room temperature. Primary antibodies were used per the manufacturer’s instructions and added to 5% non-fat dry milk in 1× TBST with shaking overnight at 4 °C. Blots were washed 3 times with 1× TBST shaking for 10 min each. HRP-conjugated secondary antibody was diluted at 1:5000 in 5% non-fat dry milk in 1× TBST and incubated for 2 h at room temperature with gentle rocking. Blots were washed 3 times with 1× TBST shaking for 10 min each. The chemiluminescence reagent (Cat#RPN2232, Amersham Biosciences, Amersham, UK) was added to blots for 1 min before capturing images on the iBright imager (Thermo Fisher Scientific). Densitometry was performed in Fiji (RRID: SCR_002285, ImageJ2) [52].

### 4.7. Immunofluorescence

Cells were plated to ~80% confluency 48 h before use in an Ibidi 8-well chamber slide with ibiTreat (200 µL/well). HBSS + Ca^2+^ was added to all cells for 1 h and used as the control. If treated with P2Y2i, the drug or DMSO (0.1%) was added to the cells for 10 min at 10 µM in the HBSS + Ca^2+^. At various points, 10 µM ATP or control ddH_2_O was added to the HBSS + Ca^2+^. Cells were fixed using 3.7% formaldehyde diluted in 1× phosphate-buffered saline (PBS) for 10 min and, henceforth, protected from light. The fixed cells were washed twice with PBS for 5 min each before WGA-Alexa 488 conjugate (Invitrogen, Cat#W11261) staining following the manufacturer’s protocol for labeling pre-fixed cells. Next, the cells were permeabilized using 0.25% Triton-X100 (USB, Cat#9002-93-1, Cleveland, OH, USA) in PBS for 10 min, followed by 2 more washes of PBS. Phalloidin (Invitrogen, Cat#A30107) was diluted in PBS at 1:1000; for nuclear co-staining, a 1:1000 dilution of Hoechst (Millipore-Sigma, Cat#33258) was added to the solution. Cells were stained for 1 h at room temperature with gentle rocking before being washed with PBS 3 times at 10 min each. A final wash with ultra-pure double-distilled water was performed before drying, fixing with Fluoromount-G (Invitrogen, Cat# 00-4958-02), and storing at 4 °C. Fixed slides were imaged on an Olympus FV-1000 confocal microscope (Olympus Scientific Solutions, Center Valley, PA, USA) at 60× magnification (oil, NA = 1.42).

### 4.8. 3-Dimensional Cell Dissemination Assay

Cell Aggregation and Spheroid Formation (Days 0–3): Confluent cells were washed with PBS and treated with 0.25% Trypsin to detach the cells. The cells were then counted and diluted to 1 × 10^6^ cells/mL based on the total volume needed (100 µL per well) before centrifugation at 1000 RPM for 5 min. Spheroid formation media were made in a 3:1 ratio of cell culture media/MethoCult H4100 (Cat#04100, StemCell Technologies, Cambridge, MA, USA), and the total volume was calculated at 100 µL per well. Cell pellets were resuspended in spheroid formation media and plated 100 µL per well in a round bottom low-attached 96-well plate (Cat#7007, Corning). The whole 96-well plate was then centrifuged to allow the cells to aggregate at 1400 RPM for 7 min; then, the plate was rotated for repeated centrifugation. Cell aggregates are allowed to grow undisturbed for 72 h to form spheroids at 37 °C, 5% CO_2_, and 95% humidity.

Spheroid Embedment (day 3): Collagen I solution (3 mL total) was prepared on ice following the protocol from Doyle, A. D., 2016 [53]. Collagen Type I was purchased from Corning (Cat#354236) and prepared at 2 mg/mL. The reconstitution buffer was prepared, and NaOH was added to balance the pH, both following the protocol. Chilled cell culture media were used to make up the volume to 3 mL. Each spheroid was gently transferred from the round bottom 96-well plate in a 1 mL pipette tip to a Petri dish for isolation from the MethoCult H4100 mixture. In total, 100 µL of cold collagen I solution was added to a flat bottom low-attach 96-well plate and pre-warmed to 37 °C on a heat block within the cell culture hood. The collagen solution was allowed to partially polymerize for 2 min before collecting the spheroid in the collagen solution and adding this to the middle of the well. Spheroids were allowed to become embedded in collagen for at least 2 min at 37 °C. These steps were repeated for each spheroid; then, the whole plate was incubated for 30 min at 37 °C, 5% CO_2_, and 95% humidity to allow the collagen to fully polymerize. An additional 100 µL of warm cell culture media was added on top of the gels. Spheroids and 3-D cell dissemination were imaged from day 0 to day 7 at 10× phase contrast (NA = 0.30), and image tiling (2 × 2) was implemented when needed on a Nikon Ti2-E inverted microscope. When indicated, some experimental conditions were treated with P2Y2i at 10 µM or control (DMSO 0.1%) in media and replenished every 24 h.

Image Analysis (post-experiment): Three images (*n* = 3) were first analyzed with NIS-Elements AR (Nikon) software version 6.02.03. The spheroid perimeter was manually traced, and the whole spheroid area was automatically quantified and measured in µm^2^; then the, change in the area was calculated and normalized to day 0 in GraphPad Prism 9.0.

### 4.9. Bioinformatics

The online cBioPortal (RRID: SCR_014555) database (https://www.cbioportal.org/ (accessed on 7 July 2022)) was used to test whether P2Y2 mRNA expression had an effect on disease-free survival in two different breast cancer datasets. The online Kaplan–Meier Plotter (RRID: SCR_018753) database (https://kmplot.com/analysis/ (accessed on 7 July 2022)) was used to test whether P2Y2 mRNA expression had effects on breast cancer patient clinical outcomes. The gene symbol used was P2RY2.

### 4.10. Statistics

Statistical analyses were conducted using two-tailed *t*-test, one-way ANOVA with Dunnett’s or Holm–Sidak multiple comparison post-tests, or two-way ANOVA with Geisser–Greenhouse correction and Tukey’s multiple comparisons test in GraphPad Prism 9.0 software; *p* < 0.05 was considered significant.

## Figures and Tables

**Figure 1 ijms-26-04286-f001:**
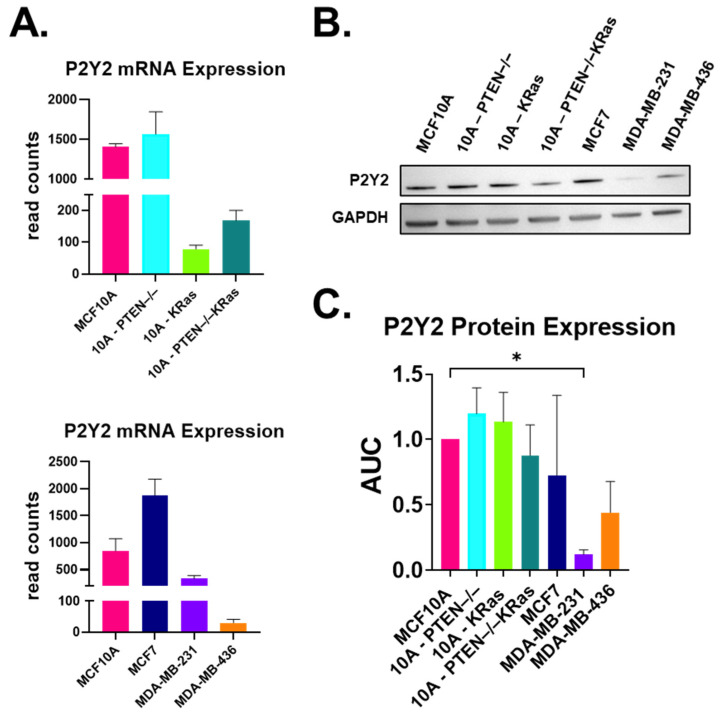
P2Y2 expression is lower in the metastatic MDA-MB-231 cell line compared to non-tumorigenic MCF10A. (**A**) mRNA sequencing results show a lower gene expression of the P2Y2 receptor in 10A-KRas, 10A-PTEN−/−KRas, MDA-MB-231, and MDA-MB-436 cells in comparison to the MCF10A control. (**B**) Representative Western blot for the P2Y2 (42 kDa) receptor across seven different cell lines shows a decrease in P2Y2 protein expression in the metastatic MDA-MB-231 cancer cell line with the GAPDH (37 kDa) loading control. (**C**) There were decreases in protein expression for the metastatic breast cancer cell lines, although the only significant difference was in MDA-MB-231 cells. Quantification of Western blots with P2Y2 expression normalized to GAPDH then compared to MCF10A using a one-way ANOVA with multiple comparisons with SD (*, *p* < 0.05) (*n* = 3).

**Figure 2 ijms-26-04286-f002:**
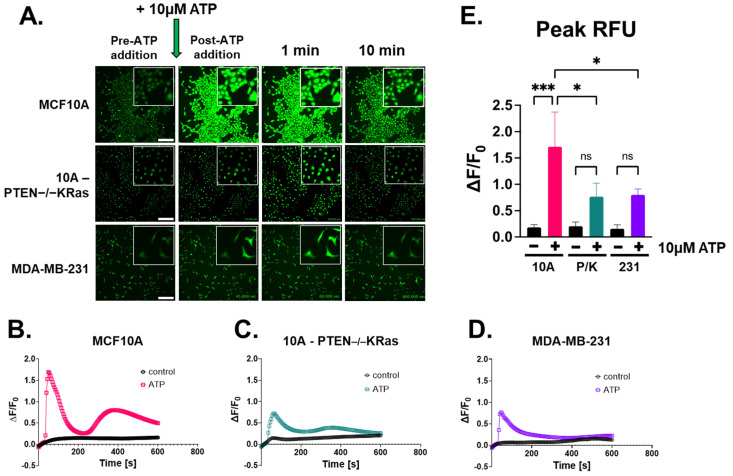
ATP, as a pseudo-mechanical stimulus, causes intracellular calcium to increase in breast epithelial cells. (**A**) The 20× epi-fluorescent time-lapsed images of MCF10A, 10A-PTEN−/−KRas, and MDA-MB-231 cells loaded with Fluo-4 AM to label calcium. In total, 10 µM ATP was added after a 30 s baseline read. Representative images of pre-ATP addition, post-addition, 1 min, and 10 min were taken (scale bar = 200 µm), and insets were taken at 4× zoom. (**B**) The graph of ∆F/F_0_ from the whole field of view showed a rapid Ca^2+^ response in MCF10A cells treated with 10 µM ATP (pink) and control (black). (**C**) 10A-PTEN−/−KRas (P/K) cells saw a suppressed intracellular Ca^2+^ response (teal) compared to control (black). (**D**) Similarly, MDA-MB-231 cells also had an inhibited Ca^2+^ response after ATP stimulation (purple) compared to control (black). (**E**) Peak RFU with one-way ANOVA and multiple comparisons performed on graphs from each cell line with SD. Colors coordinate with (**B**–**D**) (***, *p* < 0.001, *, *p* < 0.05, ns, not significant) (all graphs *n* = 3).

**Figure 3 ijms-26-04286-f003:**
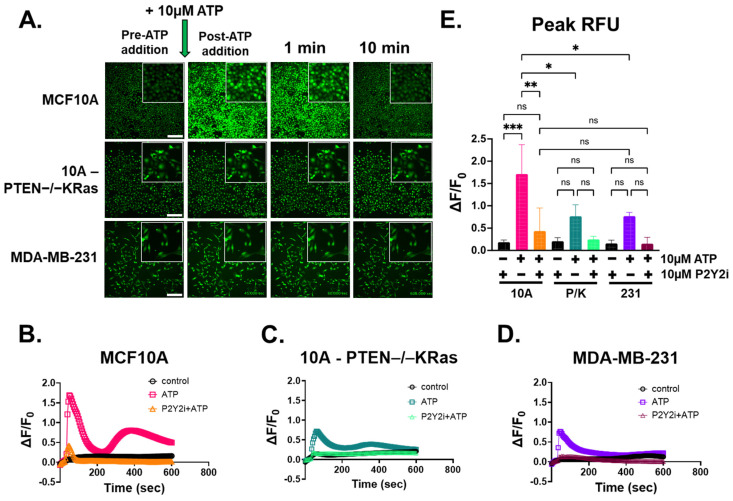
The P2Y2 antagonist, AR-C118925XX, interrupts the calcium signal stimulated by ATP. (**A**) Epi-fluorescent images of MCF10A, 10A-PTEN−/−KRas, and MDA-MB-231 cells loaded with Fluo-4 and treated with 10 µM P2Y2i for 10 min before imaging. Representative frames of pre-ATP addition, post-addition, 1 min, 5 min, and 10 min were taken from time-lapsed images (scale bar = 200 µm), and insets were taken at 4× zoom. (**B**) The graph of ∆F/F_0_ from the whole field of view showed a suppressed Ca^2+^ response in MCF10A cells treated with 10 µM P2Y2i before ATP stimulation (orange) compared to ATP with no antagonist (pink) and control (black). (**C**) 10A-PTEN−/−KRas cells saw no significant change from the control (black) with P2Y2i treatment (light green). (**D**) MDA-MB-231 cells also had no significant Ca^2+^ changes with P2Y2i treatment (maroon). (**E**) Peak RFU with one-way ANOVA and multiple comparisons performed on graphs from each cell line with SD. Colors coordinate with (**B**–**D**) (***, *p* < 0.001. **, *p* < 0.01, *, *p* < 0.05, ns, not significant) (*n* = 3).

**Figure 4 ijms-26-04286-f004:**
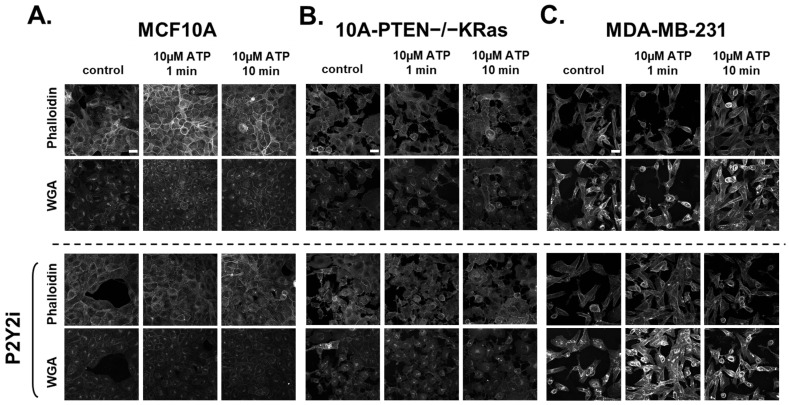
An ATP-activated increase in cytoplasmic calcium leads to rapid actin localization to cellular edges in MCF10A cells. All cells were treated with a control or 10 µM ATP for 1 min or 10 min. (**A**) Confocal images of MCF10A cells stained with phalloidin show changes in actin polymerization and localization to cellular edges and junctions after ATP stimulation. However, when using the P2Y2i, MCF10A cells show no changes in actin localization after ATP stimulation. (**B**) Confocal images of 10A-PTEN−/−KRas cells show no major changes in phalloidin staining after ATP addition with or without P2Y2i. (**C**) Confocal images of MDA-MB-231 cells also show no changes in actin localization with control or P2Y2i and ATP stimulation. Confocal 60× (oil) NA: 1.42. All images are stained with Alexa-conjugated phalloidin (nm = 647) and WGA-Alexa (nm = 488) to stain cell membranes. Representative images were taken from *n* = 3. Scale bar = 25 µm.

**Figure 5 ijms-26-04286-f005:**
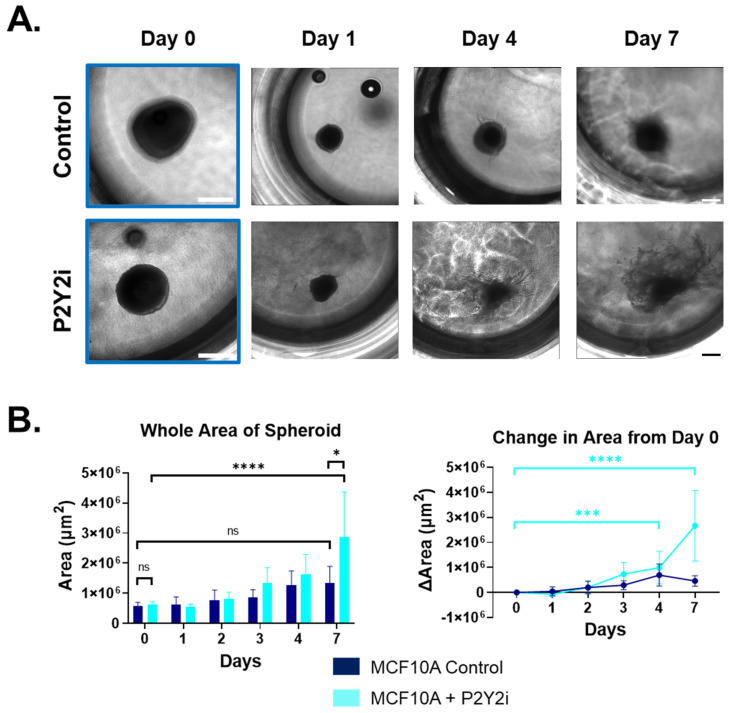
Three-dimensional cell dissemination of non-tumorigenic MCF10A cells treated with P2Y2i. Spheroids formed for 72 h were embedded into collagen solution and imaged every day for 7 days. (**A**) Representative phase-contrast images taken at 10× on day 0 (blue outline) or 10× stitched (2 × 2) spheroids treated with control or P2Y2i on days 1, 4, and 7. Yellow outlines represent 10× images unstitched (white or black scale bar = 500 µm). (**B**) Graphs showing the whole area of spheroid (**left**) and change in area (µm^2^) from day 0 to day 7 (**right**) with SD, and two-way ANOVA was performed to calculate significance (****, *p* < 0.0001, ***, *p* < 0.001 *, *p* < 0.05, ns, not significant) (representative images from *n* = 3).

**Figure 6 ijms-26-04286-f006:**
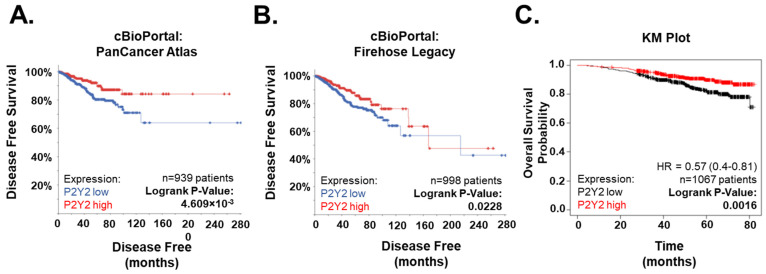
Independent survival curves for breast cancer patients based on expression of P2Y2 receptor. (**A**) Data from cBioPortal: PanCancer Atlas patients, including all subtypes of breast cancer, show that patient tumors with lower P2Y2 mRNA expression have poorer disease-free survival (*p* = 4.609 × 10^−3^). (**B**) Data from cBioPortal: Firehose Legacy patients with breast cancer and low P2Y2 mRNA expression also have smaller percentage of disease-free survival (*p* = 0.0228). (**C**) Kaplan–Meier Plot from KM Plot shows that low expression of P2Y2 receptor is correlated with poorer probability of overall survival (*p* = 0.0016).

## Data Availability

Data are provided within the manuscript or Appendix A. The datasets generated during and/or analyzed during the current study are available from the corresponding author on reasonable request. The datasets generated during and/or analyzed during the current study are available in the cBioPortal (RRID: SCR_014555) or KM Plot (RRID: SCR_018753) repository (see methods for further information). [https://www.cbioportal.org/ (accessed on 7 July 2022)] [https://kmplot.com/analysis/ (accessed on 7 July 2022)].

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
