# Peer review of "Disruption of P2Y2 Signaling Promotes Breast Tumor Cell Dissemination by Reducing ATP-Dependent Calcium Elevation and Actin Localization to Cell Junctions"

_ijms, 2025, doi:10.3390/ijms26094286_

Round 1
Reviewer 1 Report
Comments and Suggestions for Authors
Summary
The authors present a comprehensive analysis of P2Y2 signaling in breast cancer, offering valuable insights into breast epithelial cell signaling and its alterations in metastatic conditions. The manuscript is well-written and clearly articulates the aim and relevance of the study. Only minor revisions are necessary before the manuscript can be considered for acceptance.
Title
Please ensure the title is formatted according to MDPI’s guidelines by capitalizing all relevant words.
Abstract
- Lines 24–27: The background section could be made more concise, ideally reduced to 2–3 lines, to allow greater emphasis on the study's key findings.
- A brief mention of the main methods used should be included in the abstract.
- Lines 30–34: Please indicate the actual rate or percentage increase mentioned here for clarity.
Introduction
- Lines 116–132 should be condensed to 2–3 lines, focusing solely on clearly stating the aim of the study.
Materials and Methods
- The sequences of the P2Y2 primers used should be provided to ensure reproducibility.
- Please indicate the statistical software used for data analysis.
Reviewer 2 Report
Comments and Suggestions for Authors
The study is a well-written and explores the critical role of P2Y2 signaling in breast epithelial cell behavior, particularly in the context of tumorigenesis and cell dissemination. The study presents robust data linking reduced P2Y2 expression to decreased calcium signaling and actin remodeling in metastatic breast cancer models. However, a few changes could improve the study.
3-Dimensional Cell Dissemination Assay
-While the qualitative imaging is compelling, the quantitative analysis of spheroid dissemination is insufficiently detailed. Please clarify.
Discussion
- Could the authors of the study stratify P2Y2 expression and survival data in TNBC and non-TNBC? Briefly discuss about it.
Round 2
Reviewer 1 Report
Comments and Suggestions for Authors
Authors addressed all the Reviewer's comments.
Reviewer 2 Report
Comments and Suggestions for Authors
Accept